# Dependence of Exchange Bias on Interparticle Interactions in Co/CoO Core/Shell Nanostructures

**DOI:** 10.3390/nano12183159

**Published:** 2022-09-12

**Authors:** Suchandra Goswami, Pushpendra Gupta, Sagarika Nayak, Subhankar Bedanta, Òscar Iglesias, Manashi Chakraborty, Debajyoti De

**Affiliations:** 1Material Science Research Lab, The Neotia University, Sarisa, D.H. Road, 24 Pgs (South), Sarisha 743368, West Bengal, India; 2Laboratory for Nanomagnetism and Magnetic Materials (LNMM), School of Physical Sciences, National Institute of Science Education and Research (NISER), An OCC of Homi Bhabha National Institute (HBNI), Jatni 752050, India; 3Center for Interdisciplinary Sciences (CIS), National Institute of Science Education and Research (NISER), An OCC of Homi Bhabha National Institute (HBNI), Jatni 752050, India; 4Department Física de la Matèria Condensada and IN2UB, Facultat de Física, Universitat de Barcelona, Av. Diagonal 647, 08028 Barcelona, Spain; 5Department of Physics, Sukumar Sengupta Mahavidyalaya, State Highway 7, Keshpur, Paschim Medinipur 721150, India

**Keywords:** nanomaterials, core/shell nanostructures, exchange bias, interparticle interactions

## Abstract

This article reports the dependence of exchange bias (EB) effect on interparticle interactions in nanocrystalline Co/CoO core/shell structures, synthesized using the conventional sol-gel technique. Analysis via powder X-Ray diffraction (PXRD) studies and transmission electron microscope (TEM) images confirm the presence of crystalline phases of core/shell Co/CoO with average particle size ≈ 18 nm. Volume fraction (φ) is varied (from 20% to 1%) by the introduction of a stoichiometric amount of non-magnetic amorphous silica matrix (SiO${}_{2}$) which leads to a change in interparticle interaction (separation). The influence of exchange and dipolar interactions on the EB effect, caused by the variation in interparticle interaction (separation) is studied for a series of Co/CoO core/shell nanoparticle systems. Studies of thermal variation of magnetization (M−T) and magnetic hysteresis loops (M−H) for the series point towards strong dependence of magnetic properties on dipolar interaction in concentrated assemblies whereas individual nanoparticle response is dominant in isolated nanoparticle systems. The analysis of the EB effect reveals a monotonic increase of coercivity ($H_{C}$) and EB field ($H_{E}$) with increasing volume fraction. When the nanoparticles are close enough and the interparticle interaction is significant, collective behavior leads to an increase in the effective antiferromagnetic (AFM) CoO shell thickness which results in high $H_{C}$ and $H_{E}$. Moreover, in concentrated assemblies, the dipolar field superposes to the local exchange field and enhances the EB effect contributing as an additional source of unidirectional anisotropy.

## 1. Introduction

Nanoscience and nanotechnology fundamentally emerge from the manipulation of matter at the nanoscale and the curiosity to understand various properties of matter at the atomic level. In nanoparticle assemblies, parameters such as size [1], surface structure [2], shape [3], agglomeration [4] or interparticle interactions [5] often influence their properties. At the same time, they lead to the emergence of enriched physico-chemical properties, which distinguish them from their bulk counterparts. Among different classes of nanomaterials, core/shell structures are fundamentally interesting because of carrying two different physico-chemical properties in one single particle at the nanoscale [6]. Though primarily core/shell structures were synthesized to protect and stabilize the metallic core [7], advances in materials fabrication and synthesis have made core/shell structures potential candidates for a myriad of new applications including targeted drug delivery [8], biomedical sensors [9], enhanced electronic properties [10] or EB effect [6]. If the core and shell are composed of two materials with different magnetic orders, the interfacial region may experience a structural modification due to differences in the crystalline structures of both regions as well as a competition between the different magnetic orders favored at the core and shell. This leads to the phenomenon known as EB effect [11] which was first reported by Meiklejhon and Bean. The EB effect has been explained via unidirectional exchange anisotropy in Co/CoO (ferromagnetic (FM)—antiferromagnetic (AFM)) core/shell nanoparticle system [12]. Since then, this has been intensely studied in many magnetically coupled systems such as FM/FiM (ferrimagnetic) [13], AFM/FiM [14], AFM/SG (spin-glass) [15] or FiM/SG [16].

Despite intensive experimental research in the field, there are phenomena like spontaneous EB [17], EB in alloys and compounds [18,19], EB in single phase magnetically inhomogeneous materials [20] or EB in thin films [21,22] that are still drawing attention because of the urge to understand new fundamental physics. Furthermore, EB reveals a wide range of potential applications in recording media to overcome the superparamagnetic (SPM) limit [23], field sensors [24], read heads [25], giant magnetoresistance (GMR) based devices [26] and many more as well. In this regard, tuning EB-related properties by controlling variation of size [27], the thickness of core and shell [11], interparticle interactions [28] and microscopic structure of the interface of any magnetically inhomogeneous system [20] might add significant value in several application-oriented phenomena. Attempts have been made to understand the underlying physics of the EB mechanism when the variation of shape [29], size [14], surface composition [30], core to shell diameter ratio come into play [6], in a core/shell nanostructure. Recently, we have reported correlating experimental findings and atomistic Monte Carlo (MC) simulations showing that the variation of core and shell thickness of Co-Co${}_{3}$O${}_{4}$ nanostructure leads to systematic changes in the EB effect [6].

Via controlled oxidation on the surface of transition metal nanoparticles, a shell of metal oxide (generally AFM/FiM in nature) can be formed to prepare a metal/metal oxide core/shell structures [31,32]. Co/CoO is the most studied core/shell nanostructure because of its large interface energy with a high EB field ($H_{E}\approx$100 mT) compared to others [11,33]. Besides potential technological applications of Co-based nanoparticles in information storage, magnetic fluids, catalysis, etc., low crystal anisotropy of Co is favorable for FM/AFM Co/CoO as a model system for EB studies [34]. Additionally, because of high AFM N éel temperature ($T_{N}$≈285 K) of CoO, followed by a wide temperature range of EB effect [35], nanostructures of the same have been revisited by researchers to understand different phenomenological models related to EB. In particular, recent studies have reported how the shell thickness [35], the degree of oxidation of the shell [36], the core to shell diameter ratio [11], and the degree of dilution within non-magnetic matrix [37] affect EB in Co/CoO nanoparticles.

When particles are in close proximity, the magnetic properties of the nanoparticle assembly are mainly governed by exchange interactions between the surfaces in contact. Instead, long-range dipolar interactions between the macroscopic magnetic moments of the individual particles can be relevant over a wide range of interparticle separations. As a consequence, it is expected that the thermal and field dependence of the magnetization of the assembly may be very different from that of an individual nanoparticle, giving rise to a variety of behaviors such as superparamagnetism [38,39], superspin-glass [40], and superferromagnetism [39,41] among others. Some progress has been made in recent studies of frozen ferrofluids [42], granular nanoparticles [43] and diluted magnetic systems [44].

However, to the best of our knowledge, a systematic study to understand the effect of variation of interparticle interactions on EB mechanism in core/shell nanostructure keeping the core and shell diameters fixed, has not been yet reported. In the present article, we aim to study how interparticle interactions among core/shell nanoparticles can influence the phenomenology of EB. Interparticle interactions may be varied by changing the volume fraction (φ), but a high level of dilution is required for a comparative study of their effects on the magnetic properties [5]. Herein, we have performed a systematic and detailed study of the EB effect by changing the interparticle separation of Co/CoO core/shell nanoparticle system in seven different batches. Experimental findings point out that collective magnetization is hindered by the increase in the separation of particles via decreasing volume fraction, which leads to a monotonic reduction of coercivity ($H_{C}$) and EB field ($H_{E}$).

## 2. Experimental

Core/shell nanocrystalline Co/CoO is derived via controlled oxidation-reduction from nanocrystalline Co which is synthesized using conventional sol-gel technique [2]. To begin with, Co metal powder (Aldrich, 99.99% pure) is dissolved in a minimum quantity of 37% nitric acid and vigorously stirred in a magnetic rotor for 12 h until the solution becomes transparent. A stoichiometric amount of citric acid is added to the solution and homogenized for 6 h to obtain a transparent reddish solution. This ensures that all the metal ions are mixed at the atomic scale [45]. The solution is dried very slowly at room temperature for a few days. To increase the evaporation rate, the solution is kept inside a vacuum oven at 50 °C for a few days. After the solution has transformed into a gel-like state, it is heated at 100 °C to form a cake. This cake is then ground and heated at 600 °C for 6 h in presence of a continuous flow of Ar-H${}_{2}$ gas (95% Ar and 5% H${}_{2}$). Thus, Co nanoparticles are obtained. Now, as oxidation will start from the surface, the oxide shell is created via controlled oxidation at ambient temperature and the procedure is standardized after repetitive trials. The as-synthesized Co nanoparticle is then heated in the open air at 200 °C for 6 min to form an oxide shell over the Co nanoparticle core. The shell consists of both CoO and Co${}_{3}$O${}_{4}$ phases (as evident from the XRD pattern described later). To synthesize desired Co/CoO, this as-synthesized core/shell sample is annealed at 250 °C in a continuous flow of the same reducing gas (95% Ar and 5% H${}_{2}$) for 1 h [32]. This reduces excess oxygen from Co${}_{3}$O${}_{4}$ and retains the only stable CoO shell on the surface of the Co core. Thus, the Co/CoO core/shell nanoparticle assemblies are obtained.

Variation of interparticle interactions is introduced via incorporation of non-magnetic amorphous silica matrix (SiO${}_{2}$) into Co/CoO core/shell nanoparticles. The stoichiometric amount of SiO${}_{2}$ is added to the as-synthesized sample and homogenized. This reduces the volume fraction and thus increases the interparticle separation leading to the simultaneous change in the interparticle interaction. An excess amount of silica is added via mechanical grinding which results in a series of samples with volume fractions (φ) 20%, 15%, 10%, 7.5%, 5%, 1% and 0.1%. This provides the platform to study the effect of variation of interparticle interactions on magnetic properties. From now on, different samples of Co/CoO embedded in SiO${}_{2}$ matrix with a variation of interparticle interactions will be designated as CS-20, CS-15, CS-10, CS-7.5, CS-5, CS-1 and CS-0.1. For comparison, a pure Co/CoO core/shell nanoparticle is named CS-100. Figure 1 demonstrates a schematic diagram of changing volume fraction.

The structural characterization of the sample has been performed via Powder X-Ray diffraction (PXRD) pattern, recorded in a Bruker D8 Advanced Diffractometer using Cu Kα (λ = 1.54184 Å) radiation source with a scan speed of 0.02°/4 s. The actual shape, grain size and morphology of the nanoparticles are assessed by Transmission Electron Microscope (TEM), equipped with an energy dispersive X-ray spectrometer (JEOL TEM, JEM-F200). Temperature-dependent dc magnetization measurements are performed via a commercial SQUID magnetometer (MPMS-3). In the zero-field cooled (ZFC) protocol the sample is cooled in zero fields and the magnetization is recorded in a static magnetic field during the heating cycle. In the field-cooled (FC) protocol, sample is cooled in presence of a static magnetic field and magnetization measurements are performed in cooling mode.

## 3. Results and Discussions

### 3.1. Structural Characterization

PXRD pattern of as-synthesized primary sample Co, intermediate sample Co/(Co${}_{3}$O${}_{4}$ + CoO) and final product Co/CoO are recorded in the range of 30°–80°, at 300 K and are depicted in Figure 2a–c. An elaborate Rietveld refinement has been performed on the diffraction patterns using MAUD (materials analysis using diffraction) software, considering the face-centered Fm3m space group for Co, CoO and Fd3m for Co${}_{3}$O${}_{4}$. A close match between the experimental data and the computed curves is noticed, as indicated by the difference plots at the bottom of Figure 2a–c. Figure 2a corresponds to Co nanoparticle only whereas Figure 2b represents the XRD pattern of as-synthesized Co/(Co${}_{3}$O${}_{4}$ + CoO) developed after oxidation treatment of Co nanoparticles in the open air. Co/(Co${}_{3}$O${}_{4}$ + CoO) demonstrates that the characteristic peaks of Co, Co${}_{3}$O${}_{4}$ and CoO, are in accordance with the JCPDS data (15-0806), (43-1003) and (43-1004), respectively. The weight percentage of different compositions present in this intermediate sample is Co:Co${}_{3}$O${}_{4}$:CoO ≈ 26:68:6, as evident from the refinement results. Figure 2c shows the peak positions corresponding to Co and CoO phases only. Proper indexing of all the peaks in Figure 2c rules out the possibility of the presence of any other secondary phases or impurity in the final product Co/CoO. Refinement also suggests that the weight percentage of Co:CoO ≈ 20:80. This suggests that controlled oxidation-reduction converts Co${}_{3}$O${}_{4}$ into stable CoO phase [32]. Vertical bars at the bottom of Figure 2a–c in three different colors correspond to different phases (Co, Co${}_{3}$O${}_{4}$ and CoO). Information extracted from the Rietveld refinement such as, lattice parameters, atomic positions, refined parameters ($R_{p}$, $R_{wp}$ and $\chi^{2}$ as shown in Table 1), bond angles are in acceptable range and are in close agreement with recent results [46,47,48]. An increase in average crystallite sizes (*D*) of the three samples, as determined by modified Scherrer’s formula [49,50] from the corresponding PXRD patterns, commensurate with the heat treatment (see Table 1).

To investigate the actual size and morphology of the samples and their procedural changes with synthesis, TEM micrographs are taken for all three samples. Figure 3a,b,e,f,i,j depict spherical nanoparticles for Co, Co/(Co${}_{3}$O${}_{4}$ + CoO) and Co/CoO, respectively. Insets of Figure 3a,e,i represent the histograms of particle size distributions of the samples, fitted with log-normal distribution function. The results of the fitted average particle sizes are found to be ≈12 nm for Co ($\sigma_{log}$ = 0.15), ≈17 nm for Co/(Co${}_{3}$O${}_{4}$ + CoO) ($\sigma_{log}$ = 0.18) and ≈18 nm for Co/CoO ($\sigma_{log}$ = 0.23). Here too, the gradual increase in particle sizes supports the findings of the PXRD studies and is in accordance with the heat treatment. Figure 3c,g,k represent the high resolution (HR) TEM images of the samples revealing the formation of high crystallinity up to the edges of the particles. Lattice plane spacing of (111) plane for Co, (311) for Co${}_{3}$O${}_{4}$ and (111) for CoO are observed in different HR-TEM images which correspond to the formation of core/shell structure. Selected area electron diffraction (SAED) patterns of the nanoparticles are presented in Figure 3d,h,l where planes corresponding to Co, Co${}_{3}$O${}_{4}$ and CoO are observed. Findings support the PXRD studies and no impurity plane is noted. The results obtained from two different techniques replicate almost similar results, indicating the purity of the samples. However, the little deviation in crystallite sizes obtained from PXRD and TEM are in accordance with recent reports [2,20]. Subsequently, the duly characterized core/shell Co/CoO nanoparticles were submitted to the mechanical grinding treatment with SiO${}_{2}$ explained in Section 2 to obtain the samples with different volume fractions.

### 3.2. Magnetic Characterization

#### 3.2.1. ZFC-FC Thermal Dependence

To understand the effect of interparticle interactions on the magnetic behavior of as-synthesized Co/CoO nanoparticles with different volume fractions, thermal variation of magnetization (M−T) in ZFC and FC modes in an applied field of 10 mT field were measured and the results are shown in Figure 4 for CS-10, CS-5 and CS-1 samples as representative of the series. The general trend of M−T curves is similar for the three samples, showing irreversibility up to the maximum measured temperature of 320 K [51] and suggesting that the blocking temperature is above this value, which is reasonable, given the relatively big nanoparticle sizes. All the curves decrease monotonously below 320 K. However, ZFC curves show a subtle but noticeable anomaly at ≈290 K, which corresponds to the Néel temperature ($T_{N}$) of the AFM CoO shell [11,34,35] and explains the steeper decrease of the magnetization below this temperature. No anomaly is observed at the ordering temperature of Co${}_{3}$O${}_{4}$ (≈40 K), confirming the absence of this phase as also evidenced by PXRD. We notice a sudden increase in magnetization below 12 K for all the samples, generally known as Curie tail-like behavior [52], that is usually observed in systems with broken spin chains or paramagnetic-like impurities [53]. Recent reports suggest that the presence of oxygen atoms in the CoO shell generates holes that break the infinite lattice chain and thus can lead to the low-temperature uptail [54]. The Curie-tail is more pronounced as the volume fraction increases, indicating that upon dilution, collective magnetic effects diminish leading to a reduction of the Curie-tail. Curves of samples with increasing φ in Figure 4 display a progressive decrease of the magnitude of the magnetization and that is the first evidence that interparticle interactions change the magnetic behavior of the core/shell nanoparticles.

#### 3.2.2. ZFC Hysteresis Loops

In order to study the influence of interactions on the reversal by a magnetic field, M−H hysteresis loops for all the seven samples with different φ have been measured in between ±5 T magnetic fields after cooling the samples from 300 K to 4 K in zero field (ZFC), as displayed in Figure 5. For the most diluted samples (CS-0.1, CS-1), the loops do not saturate even at 5 T. The observed high field quasi linear response characteristic of an SPM is a typical signature of a disordered system. In these cases, the response to a magnetic field as reflected in the loops must come from intrinsic properties of an individual core/shell particles. Surface spins may present frustration due to broken links or lack of coordination, and increased surface anisotropy [55], which distinguish them from those in the inner regions of the shell that are pinned by the coupling to core spins and do not contribute to the magnetic response [34]. Therefore, the absence of saturation at low temperatures with lower φ can be attributed to the progressive alignment of the outermost layer of the AFM shell spins towards the core magnetization.

However, this behavior changes to M−H loops that saturate towards lower values of the magnetization as the interparticle separation is decreased. This can be seen in Figure 5 for samples CS-5 to CS-20. In the case of CS-20, the magnetization saturates already at 1 T to $M_{S}\approx 27$ Am${}^{2}$/kg, whereas for the most diluted system CS-0.1, $M_{S}\approx$ 111 Am${}^{2}$/kg at 5 T field. As φ increases, the mean interparticle separation decreases and dipolar interactions become more relevant. Thus, in the more concentrated assemblies, the magnetic behavior is dominated by collective effects induced by dipolar interactions of the FM Co cores, the M−H loops display the typical shape of a superferromagnetic (SFM) system [39] and the SPM contribution coming from the individual nanoparticle response is suppressed. Assuming that dipolar interactions arise from the FM cores (neglecting the AFM shell contribution to the total magnetization), typical dipolar energies between two Co particles separated by a distance *d* twice the shell thickness can be evaluated as $E_{\mathrm{dip}}=\frac{\mu_{0}}{4\pi }\frac{M_{s}^{2}V^{2}}{d^{3}}\approx$ 1220 K, to be compared with the typical anisotropy energy $E_{\mathrm{ani}}=KV\approx$ 3030 K, which qualifies our samples with higher volume fraction as governed by collective dipolar behavior [56,57]. The overall behavior is consistent with the changes in the magnetization in the low-temperature region of the M−T curves shown in Figure 4.

#### 3.2.3. Exchange Bias

The influence of interactions on the EB effect was evidenced by recording hysteresis loops after cooling in an applied field $H_{\mathrm{cool}}$ = 1 T. The central portions of the loops are shown by red dashed lines in Figure 6a–h and ZFC loops are displayed in the same figure as black solid lines. For comparison, the M−H loop of CS-100, (Co/CoO without SiO${}_{2}$) is also included and insets in the corresponding panels show the M−H loops in full scale. All loops after FC show a notable shift contrary to the cooling field direction as indicated by the arrows, with a concomitant increase in the coercive field with respect to the one measured under ZFC conditions [58]. The fact that the shift is observed even for the most diluted sample (CS-0.1) asserts the existence of a unidirectional anisotropy, originated by the freezing of the AFM shell spins, and that the resulting EB effect is induced at the individual particle level [18,19].

The EB field ($H_{E}$) and coercivity ($H_{C}$) have been determined as $H_{E}$ = $\left|H_{2}+H_{1}\right|$/2 and $H_{C}$ = $\left|(H_{2}-H_{1})\right|$/2; where $H_{1}$ and $H_{2}$ are the coercivities of the increasing and decreasing field branches, respectively, [6]. The resulting dependence of $H_{E}$ and $H_{C}$ with the volume fraction φ in the different samples are depicted in Figure 6i, which show a monotonic increase of both quantities with increasing φ, i.e., increasing interparticle interactions. In going from CS-0.1 to CS-20, $H_{C}$ and $H_{E}$ change from 119.1 to 179.3 mT and from 47.3 to 80.0 mT, respectively. Since the samples with different φ are derived from the same mother sample (CS-100) by the introduction of additional SiO${}_{2}$, it can be assumed that the contribution to the loop shift coming from the exchange coupling at the interfacial region of the individual particles is the same for all of them. Therefore, the reason for the observed notably higher values $H_{C}$, $H_{E}$ at higher concentrations must have its origin in the decrease of interparticle distance at higher φ. On one hand, it can be argued that when CoO shells come close to each other, the effective thickness of the AFM shell increases, leading to an increase in magnetic coupling and coercive field between the FM core and AFM shell as argued in [37]. However, this effect could only partially explain the observations. Recent MC simulations of a simplified macrospin model of core/shell nanoparticle assemblies [59,60] have shown that the EB field is influenced by both direct interparticle exchange and dipolar interactions, whose contributions could be separately evaluated. Simulation results were in good agreement with experimental results showing an increase of $H_{E}$ in powder samples compared to diluted ferrofluids [60,61].

We believe that, due to the higher core sizes and shell thicknesses of our samples, an additional contribution behind the EB enhancement could be the increase of the dipolar fields felt by the individual particles as the particles approach each other. In an individual core/shell nanoparticle, the loop shift is related to the local exchange field created by the uncompensated spins at the interface [62] that adds in opposite directions at the decreasing and increasing field loop branches which generate a unidirectional anisotropy. Our situation bears similarities with the dipole-induced EB model proposed in [63] to explain EB in AFM/FM thin films separated by an interface layer. When the particle is in an assembly, the dipolar field generated by the rest of the particles superposes to the local exchange field and acts as an additional source of unidirectional anisotropy that enhances the EB effect. In order to reinforce this interpretation, we will now give an estimation of typical dipolar fields that can be found in SFM samples, using an argument that was also employed to explain the shift in energy barrier distributions due to dipolar interactions [64]. Let us consider a Co core of diameter *D* = 18 nm. A rough estimate of the dipolar field felt at a distance two times the typical shell thickness *d* = 24 nm can be obtained as $H_{\mathrm{dip}}=\frac{\mu_{0}}{4\pi }\frac{M_{s}V}{d^{3}}\approx$11 mT. This is of the correct order of magnitude of the increase in 32.7 mT observed for φ = 20 in Figure 6i, if we consider that the $H_{\mathrm{dip}}$ acting on a particle receives contributions from several neighbors and that their magnetizations may not be aligned.

To study the dependence of $H_{C}$ and $H_{E}$ with cooling field ($H_{\mathrm{cool}}$), M−H loops at different fields ($H_{\mathrm{cool}}$ = 0.02, 0.05, 0.5, 1, 2.5, 5 T) were recorded in between ±5 T after cooling the sample from 300 to 4 K in FC mode. The results are shown in Figure 7a for the CS-5 sample as representative of the series. Hysteresis loops at $H_{\mathrm{cool}}$ = 1 T were also recorded for a wide temperature range (5–250 K) to study the nature of EB for CS-5 at different temperatures (refer Figure 7b). The variation of the calculated parameters $H_{E}$ and $H_{C}$ with $H_{\mathrm{cool}}$ and *T* are presented in Figure 7c,d, respectively. Both $H_{C}$ and $H_{E}$ increase with $H_{\mathrm{cool}}$ and almost become saturated near 1 T. Then, we can exclude that the observed phenomenology can be attributed to a minor loop effect, in accordance with recent reports [6,20]. Figure 7d demonstrates a monotonic decrease of $H_{C}$ and $H_{E}$ with an increase in temperature. $H_{E}$ almost vanishes at 250 K which is close to the N éel temperature of the CoO shell. The decrease of $H_{E}$ with *T* may be due to the loss of interface coupling between Co core and CoO shell caused by the increase in thermal fluctuations and the decrease in AFM anisotropy with an increase in temperature [65,66].

## 4. Conclusions

Co nanoparticles have been synthesized using a conventional sol-gel technique with an average particle size 12 nm. CoO shell has been formed over the Co nanoparticle via controlled oxidation-reduction which results in the formation of core/shell nanoparticles (Co/CoO) with average particle size 18 nm. Samples were characterized by PXRD and TEM analysis. The coexistence of Co and CoO phases is confirmed without a very regular core/shell structure. Interparticle interaction (separation) among the individual nanoparticles has been tuned by changing the volume fraction via the introduction of an additional SiO${}_{2}$ matrix. Notably, different trends in the thermal dependence of the magnetization are observed for samples with different interparticle interactions. Starting from the most concentrated sample, a gradual decrease in coercivity and increase in non-saturation tendency were obtained with the incorporation of the non-magnetic matrix from the M−H loops, demonstrating the change from SPM response due to individual nanoparticles to an SFM collective behavior as the interparticle interactions increased. Our study of EB effects has revealed that $H_{C}$, $H_{E}$ and $M_{S}$ can be tuned monotonously with the increase of the volume fraction of the core/shell nanoparticles. We have given an interpretation of the nature of the variations of $H_{C}$ and $H_{E}$ with φ, linking them to changes in the interfacial coupling of FM core and AFM shell and increase of the local fields felt by the Co cores as a consequence of increasing dipolar interactions when the distance between the nanoparticles is decreased.

It is worth noticing that the variation of EB-related parameters with interparticle interactions along with our previous findings of variation of EB with core to shell diameter ratio [6] enhances the understanding of the EB mechanism. This provides us a platform to tune or have deep control over the EB-related phenomenon and parameters of core/shell structures to explore application-oriented device fabrication. In this study, Co/CoO core/shell nanoparticle has been chosen as a representative of metal/metal oxide (FM/AFM) systems revealing the EB effect. The dependency of $H_{C}$ and $H_{E}$ with interparticle interaction reported here may be considered a general phenomenon after comparable studies with core/shell nanoparticles of similar and different compositions. Finally, we may conclude that, though the EB phenomenon was discovered about 70 years ago, there are still many other origins of the EB mechanism that differ from the conventional knowledge of the pinning mechanism at the interface between materials with different magnetic ordering. A full understanding of these new underlying mechanisms in EB assemblies will necessitate further experimental studies and will need to improve current theoretical frameworks that can incorporate collective effects due to dipolar interactions. 

## Figures and Tables

**Figure 1 nanomaterials-12-03159-f001:**
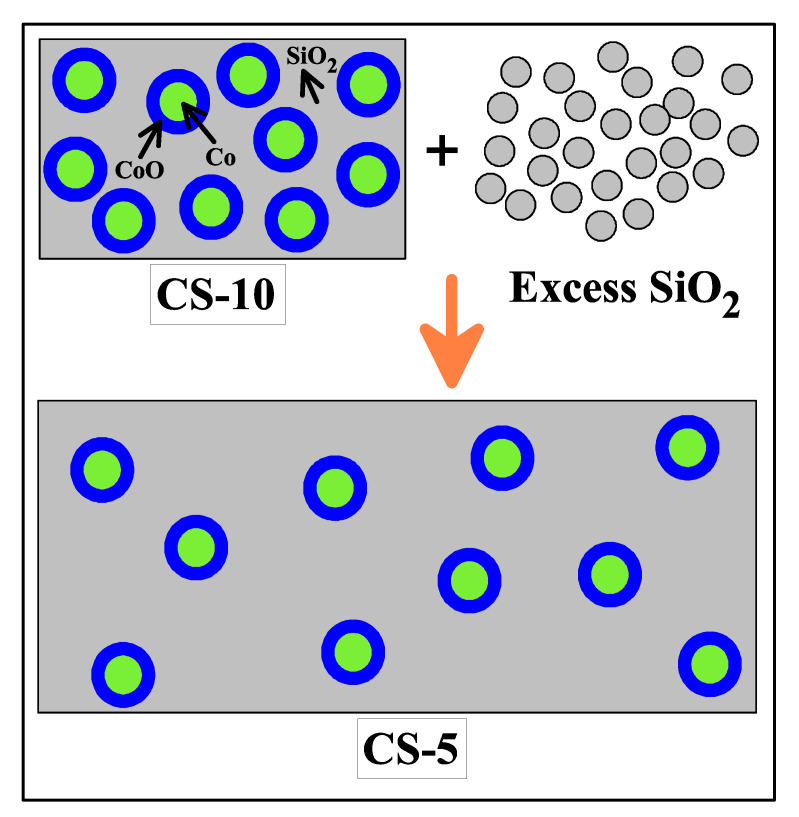
Schematic diagram representing change in volume fraction of Co/CoO nanoparticles from 10% to 5% by introducing excess silica matrix in the system.

**Figure 2 nanomaterials-12-03159-f002:**
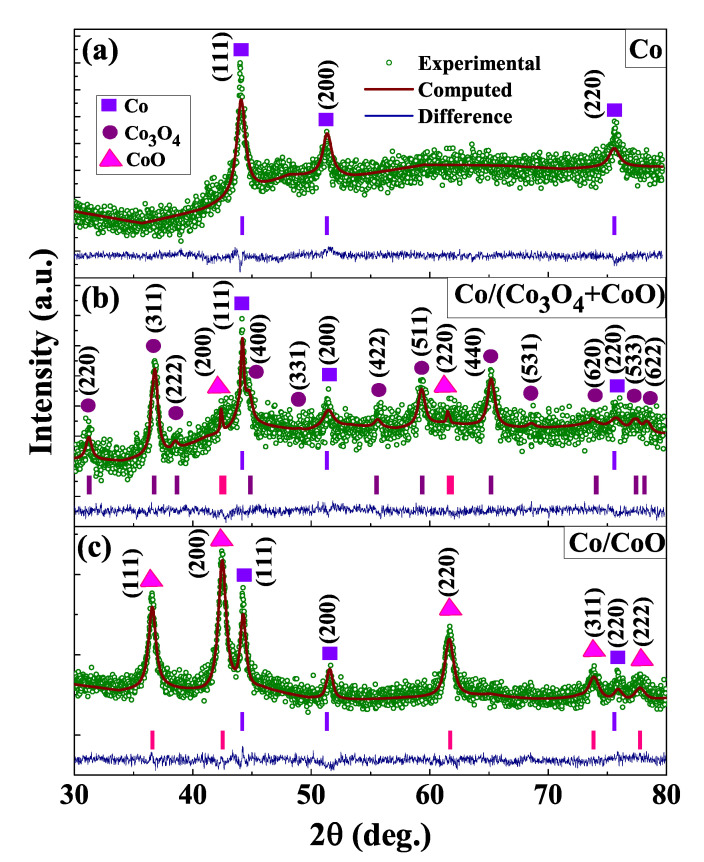
PXRD pattern of (**a**) Co nanoparticle, (**b**) Co/(Co${}_{3}$O${}_{4}$ + CoO), after heating the Co nanoparticle in open air and (**c**) Co/CoO core/shell structure. Solid continuous curves are the fits using Rietveld refinement and the lowermost plots in each panel are the residuals. Vertical bars correspond to the peak positions of the different crystalline planes.

**Figure 3 nanomaterials-12-03159-f003:**
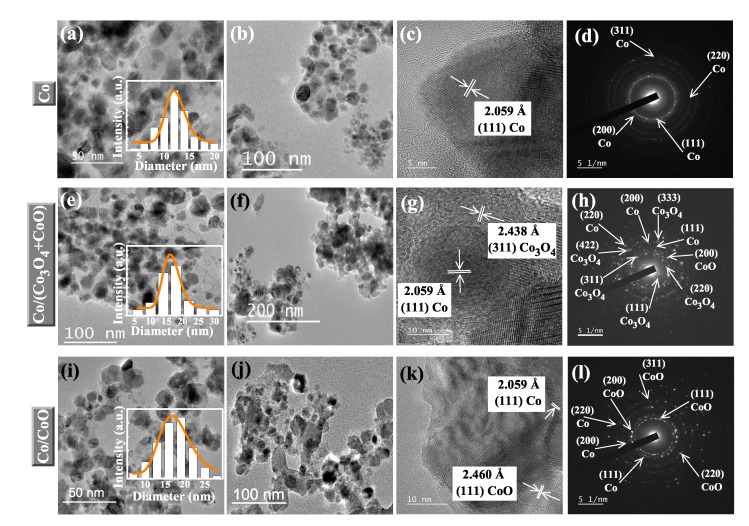
TEM images of Co (**a**,**b**), Co/(Co${}_{3}$O${}_{4}$ + CoO) (**e**,**f**)) and Co/CoO (**i**,**j**) show well dispersed particles with distribution of particle sizes. Insets of (**a**,**e**,**i**) depict the histograms of particle sizes, fitted with log-normal distribution function. (**c**,**g**,**k**) HR TEM images, highlighting plane spacings of Co, Co${}_{3}$O${}_{4}$ and CoO. (**d**,**h**,**l**) correspond to SAED pattern of the three samples, indicating different crystalline planes which are in accordance with the PXRD pattern.

**Figure 4 nanomaterials-12-03159-f004:**
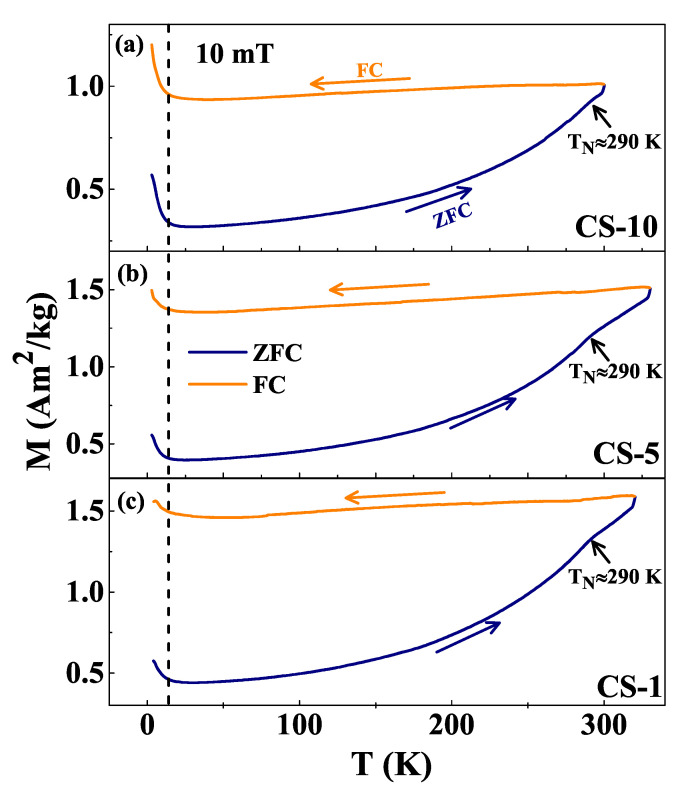
Thermal dependence of the ZFC and FC magnetization of samples (**a**) CS-10, (**b**) CS-5 and (**c**) CS-1, at 10 mT. Anomalies evident near 290 K correspond to the Néel temperature ($T_{N}$) of CoO. In the low-temperature region, a Cuire tail-like behavior is observed for all the samples.

**Figure 5 nanomaterials-12-03159-f005:**
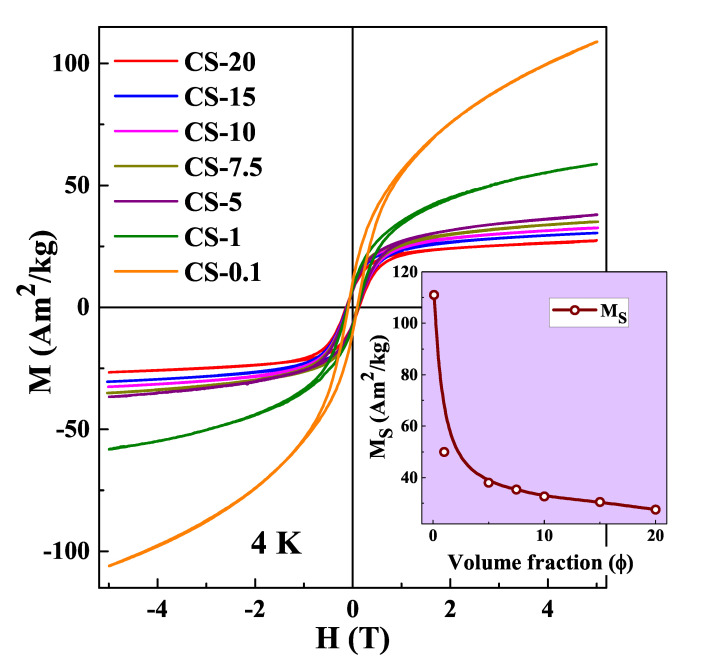
M−H loops of samples CS-20, CS-15, CS-10, CS-7.5 CS-5, CS-1, CS-0.1 measured in ZFC mode. Inset shows the variation of $M_{S}$ with volume fraction (φ).

**Figure 6 nanomaterials-12-03159-f006:**
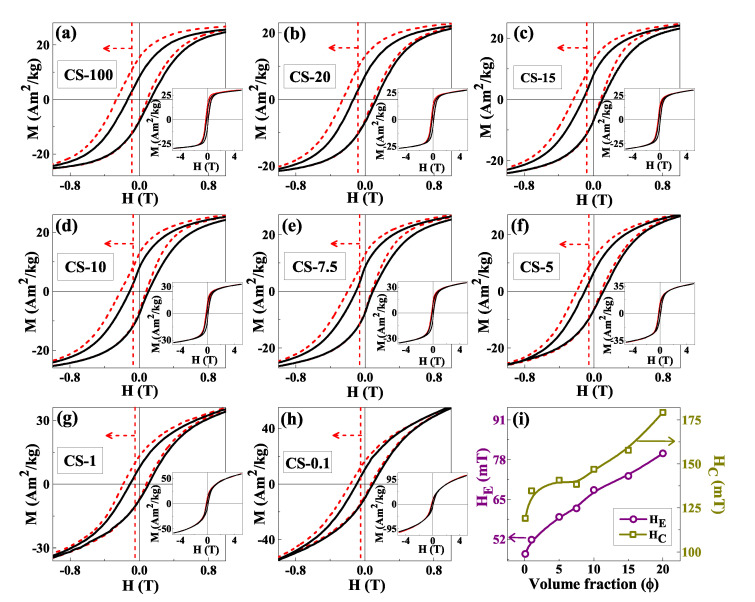
(**a**–**h**) Central portions of the ZFC (solid line) and in the FC ($H_{\mathrm{cool}}$ = 1 T, dashed line) M−H loops recorded at 4 K for CS-100, CS-20, CS-15, CS-10, CS-7.5 CS-5, CS-1 and CS-0.1. Full scale M−H loops are depicted in the corresponding insets. (**i**) Variation of coercivity ($H_{C}$) and EB field ($H_{E}$) with change in volume fraction (φ).

**Figure 7 nanomaterials-12-03159-f007:**
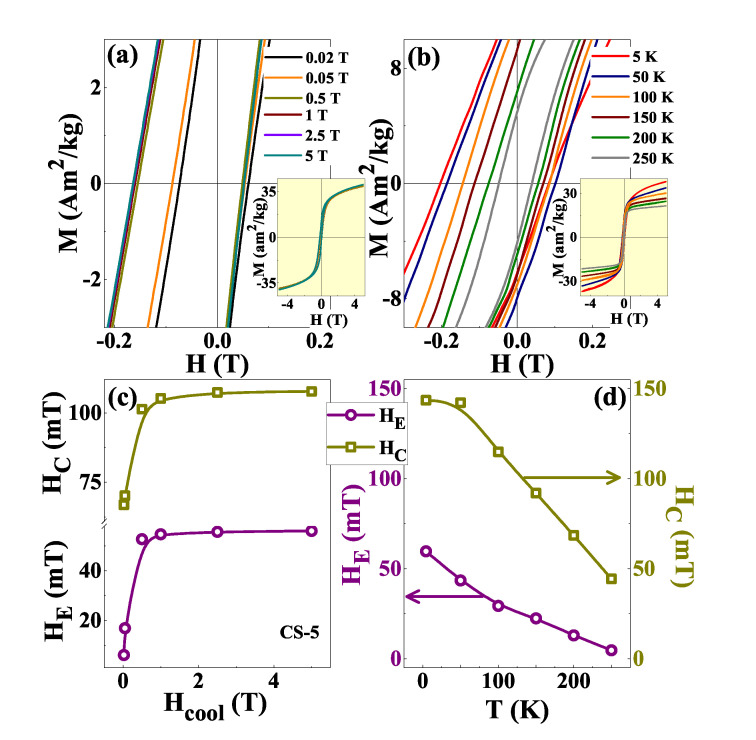
(**a**) Low field region of M−H loops of CS-5 recorded at 4 K in between ±5 T for $H_{\mathrm{cool}}$ = 0.02, 0.05, 0.5, 1, 2.5 and 5 T. Inset of (**a**) shows the corresponding M−H loops in full scale. (**b**) M−H loops of CS-5 measured at different temperatures in FC mode at $H_{\mathrm{cool}}$ = 1 T. Inset of (**b**) shows the corresponding M−H loops in full scale. (**c**,**d**) Variation of $H_{E}$ and $H_{C}$ with change in cooling field and temperature, respectively.

**Table 1 nanomaterials-12-03159-t001:** Lattice and refinement parameters.

Sample	Space	*a*	$R_{p}$	$R_{\mathrm{wp}}$	$\chi^{2}$	$D_{\mathrm{PXRD}}$	$D_{\mathrm{TEM}}$
	Group	(Å)	(%)	(%)		(nm)	(nm)
Co	Fm3m	3.558 (0.004)	1.8399	2.3351	1.1762	26.12 (0.03)	12.22 (0.09)
	Fm3m (Co)	3.551 (0.002)					
Co/(Co${}_{3}$O${}_{4}$ + CoO)	Fd3m (Co${}_{3}$O${}_{4}$)	8.093 (0.002)	1.6499	2.0905	1.0839	28.98 (0.02)	16.57 (0.27)
	Fm3m (CoO )	4.260 (0.001)					
Co/CoO	Fm3m (Co)	3.545 (0.001)	1.3173	1.6411	1.1198	30.06 (0.03)	17.54 (0.02)
	Fm3m (CoO)	4.255 (0.001)					

## Data Availability

All the data used in this article are with the corresponding authors and may be available on request.

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
