# Peer review of "Dependence of Exchange Bias on Interparticle Interactions in Co/CoO Core/Shell Nanostructures"

_nanomaterials, 2022, doi:10.3390/nano12183159_

Round 1

Reviewer 1 Report

In this study, the exchange bias effect of Co/CoO core/shell nanoparticles has been investigated. It is found that a monotonic increase of coercivity (HC ) and EB field (HE) with increasing volume fraction. However, the exchange bias phenomenon in Co/CoO nanostructure is already well known. A brief discussion of future applications of the Co/CoO nanoparticles synthesized in the present work is required.

Author Response

Reply to Referee-I

Overall remark of the Referee I: ‘In this study, the exchange bias effect of Co/CoO core/shell nanoparticles has been investigated. It is found that a monotonic increase of coercivity (HC) and EB field (HE) with increasing volume fraction. However, the exchange bias phenomenon in Co/CoO nanostructure is already well known. A brief discussion of future applications of the Co/CoO nanoparticles synthesized in the present work is required.’

  • Authors’ Reply: We thank the Referee for his/her favorable comments towards our manuscript. In our revised manuscript, we have taken care of the issues raised by the referee.

The motive of selecting Co/CoO, a well known system revealing exchange bias (EB) phenomenon has been added in the sections ‘Introduction’ and ‘Conclusion’ as well, which reads as-

…………… “Besides potential technological applications of Co-based nanoparticles in information storage, magnetic fluids, catalysis etc., low crystal anisotropy of Co is favorable for FM/AFM Co/CoO as a model system for EB studies [34]. Additionally, because of high AFM Neel temperature (TN≈285 K) of CoO, followed by wide temperature range of EB effect [35], nanostructures of the same has been revisited by researchers to understand different phenomenological models related to EB.”……….

and

………. “In this study, Co/CoO core/shell nanoparticle has been chosen as a representative of metal/metal oxide (FM/AFM) systems revealing EB effect. The dependency of HC and HE with interparticle interaction reported here, may be considered as a general phenomenon after comparable studies with core/shell nanoparticles of similar and different compositions.”……

Please see in the attachment the modified manuscript highlighting inclusion in the in the ‘Introduction’ and ‘Conclusion’ sections.

Reviewer 2 Report

In this manuscript, the authors systematically studied the dependence of exchange bias (EB) effects on ion interactions by varying the interparticle interactions with different volume fractions (φ). From my point of view, the experimental design and verification of this manuscript are sufficient and convincing. This manuscript may be considered for publication, but the following questions need to be appropriately supplemented.

1. Check the shorthand for special nouns in the text. The full name should only appear for the first time in the text (e.g. Exchange bias-EB).

2. Check the manuscript carefully for editing errors, such as “the volume fraction (ϕ) as 20%, 15%, 10%, 7.5% 5%, 1% and 0.1%.” The necessary comma is missing.

3. The author chose CS-20, CS-15, CS-10, CS-7.5, CS-5, CS-1and CS-0.1 when selecting the dilution concentration. What is the reason for the selection of such irregular concentration gradients? Will it affect the variation law of the experimental results?

4. When discussing the phenomenon that HC and HE values increased significantly at higher concentrations, the authors claimed that "We believe that, due to the higher core sizes and shell thicknesses of our samples, the reason behind the EB enhancement is the increase of the dipolar fields felt by the individual particles as the Particles approach to each other." Please add necessary references and theoretical support to further explain the reliability of this statement.

5. The conclusion needs to be simplified and revised. It must provide a very clear goal and present the core findings succinctly.

Author Response

Reply to Referee-II

Overall remark of the Referee II: ‘In this manuscript, the authors systematically studied the dependence of exchange bias (EB) effects on ion interactions by varying the interparticle interactions with different volume fractions (φ). From my point of view, the experimental design and verification of this manuscript are sufficient and convincing. This manuscript may be considered for publication, but the following questions need to be appropriately supplemented’

  • Authors’ Reply: We thank the Referee for his/her favorable comments towards our manuscript. In our revised manuscript, we have taken care of the issues raised by the referee.

Comment 1. Check the shorthand for special nouns in the text. The full name should only appear for the first time in the text (e.g. Exchange bias-EB).

  • Authors’ Reply: We thank the referee for his/her suggestion. We have corrected the manuscript accordingly and the full name is appearing for the first time and in section-heading now in the modified manuscript.

Comment 2. Check the manuscript carefully for editing errors, such as “the volume fraction (ϕ) as 20%, 15%, 10%, 7.5% 5%, 1% and 0.1%.” The necessary comma is missing.

  • Authors’ Reply: We thank the referee for going through our manuscript so carefully. We have done the necessary correction followed by proofreading from the beginning of the manuscript.

Comment 3. The author chose CS-20, CS-15, CS-10, CS-7.5, CS-5, CS-1and CS-0.1 when selecting the dilution concentration. What is the reason for the selection of such irregular concentration gradients? Will it affect the variation law of the experimental results?

  • Authors’ Reply: We thank for his/her critical question. We have stated in the manuscript that all the samples with different volume fraction (ϕ) are derived from the same mother sample (CS-100) by introduction of additional SiO2. Reaching homogeneity has been one crucial issue. For samples with volume fraction (ϕ) more than 20, it was difficult to homogenise via mechanical grinding because of very low volume of SiO2 compared to CS-100. After preparing CS-20, volume fraction was down by a factor of 5 (CS-15, CS-10, CS-5 and CS-1). Intermediate CS-7.5 was prepared and tested to understand the large fall of HE between CS-10 → CS-5 (refer Fig. 6(i)). We tried to understand the effect of negligible interparticle interaction on exchange bias phenomenon. So, a final product (CS-0.1) was prepared which is well homogenised and volume fraction / interparticle interaction is as low as practicable.

Comment 4. When discussing the phenomenon that HC and HE values increased significantly at higher concentrations, the authors claimed that "We believe that, due to the higher core sizes and shell thicknesses of our samples, the reason behind the EB enhancement is the increase of the dipolar fields felt by the individual particles as the Particles approach to each other." Please add necessary references and theoretical support to further explain the reliability of this statement.

  • Authors’ Reply: We sincerely thank the referee for his/her favourable suggestions. A theoretical calculation in support of experimental finding is already there at the end of the paragraph bearing the sentence quoted above. However, we have rephrased the paragraph with an improved explanation and added some more references to further support our claims. The new version of the paragraph reads as-

“However, this effect could only partially explain the observations. Recent MC simulations of a simplified macrospin model of core/shell nanoparticle assemblies [59, 60] have shown that the EB field is influenced by both direct interparticle exchange and dipolar interactions, whose contributions could be separately evaluated. Simulation results were in good agreement with experimental results showing an increase of HE in powder samples compared to diluted ferrofluids [60, 61].

We believe that, due to the higher core sizes and shell thicknesses of our samples, an additional contribution behind the EB enhancement could be the increase of the dipolar fields felt by the individual particles as the particles approach each other. In an individual core/shell nanoparticle, the loop shift is related to the local exchange field created by the uncompensated spins at the interface [62] that adds in opposite directions at the decreasing and increasing field loop branches which generates a unidirectional anisotropy. Our situation bears similarities with the dipole-induced EB model proposed in [63] to explain EB in AFM/FM thin films separated by an interface layer.”

Accordingly, 4 new references have been added to the literature.

  •  

Comment 5. The conclusion needs to be simplified and revised. It must provide a very clear goal and present the core findings succinctly.

  • Authors’ Reply: We thank the referee for his/her suggestion to modify the section ‘Conclusion’. We have taken care of this suggestion in the revised manuscript. The ‘Conclusion’ section has been divided into two paragraphs. Of which the former summarizes our findings and the later gives a general remark on the topic of research with a future goal.

Inclusions can be seen as highlighted in attachment.
